# CHARACTERIZING ADVERSARIAL SUBSPACES USING LOCAL INTRINSIC DIMENSIONALITY

**Xingjun Ma**[1]**, Bo Li**[2]**, Yisen Wang**[3]**, Sarah M. Erfani**[1]**, Sudanthi Wijewickrema**[1]
**Grant Schoenebeck**[4]**, Dawn Song**[2]**, Michael E. Houle**[5]**, James Bailey**[1]
[1]The University of Melbourne, Parkville, Australia
[2]University of California, Berkeley, USA
[3]Tsinghua University, Beijing, China
[4]University of Michigan, Ann Arbor, USA
[5]National Institute of Informatics, Tokyo, Japan

## ABSTRACT

Deep Neural Networks (DNNs) have recently been shown to be vulnerable against adversarial examples, which are carefully crafted instances that can mislead DNNs to make errors during prediction. To better understand such attacks, a characterization is needed of the properties of regions (the so-called 'adversarial subspaces') in which adversarial examples lie. We tackle this challenge by characterizing the dimensional properties of adversarial regions, via the use of *Local Intrinsic Dimensionality* (LID). LID assesses the space-filling capability of the region surrounding a reference example, based on the distance distribution of the example to its neighbors. We first provide explanations about how adversarial perturbation can affect the LID characteristic of adversarial regions, and then show empirically that LID characteristics can facilitate the distinction of adversarial examples generated using state-of-the-art attacks. As a proof-of-concept, we show that a potential application of LID is to distinguish adversarial examples, and the preliminary results show that it can outperform several state-of-the-art detection measures by large margins for five attack strategies considered in this paper across three benchmark datasets . Our analysis of the LID characteristic for adversarial regions not only motivates new directions of effective adversarial defense, but also opens up more challenges for developing new attacks to better understand the vulnerabilities of DNNs.

## 1 INTRODUCTION

Deep Neural Networks (DNNs) are highly expressive models that have achieved state-of-the-art performance on a wide range of complex problems, such as speech recognition (Hinton et al., 2012) and image classification (Krizhevsky et al., 2012). However, recent studies have found that DNNs can be compromised by adversarial examples (Szegedy et al., 2013; Goodfellow et al., 2014; Nguyen et al., 2015). These intentionally-perturbed inputs can induce the network to make incorrect predictions at test time with high confidence, even when the examples are generated using different networks (Liu et al., 2016; Carlini & Wagner, 2017b; Papernot et al., 2016b). The amount of perturbation required is often small, and (in the case of images) imperceptible to human observers. This undesirable property of deep networks has become a major security concern in real-world applications of DNNs, such as self-driving cars and identity recognition (Evtimov et al., 2017; Sharif et al., 2016). In this paper, we aim to further understand adversarial attacks by characterizing the regions within which adversarial examples reside.

Each adversarial example can be regarded as being surrounded by a connected region of the domain (the 'adversarial region' or 'adversarial subspace') within which all points subvert the classifier in a similar way. Adversarial regions can be defined not only in the input space, but also with respect to the activation space of different DNN layers (Szegedy et al., 2013). Developing an understanding of the properties of adversarial regions is a key requirement for adversarial defense. Under the assumption that data can be modeled in terms of collections of manifolds, several works have attempted to

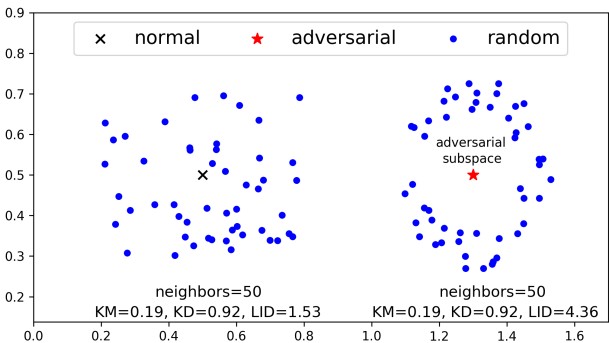

Figure 1: This example shows how density measures can fail to characterize the spatial properties of adversarial regions. The Gaussian kernel with bandwidth 0.2 is used for KD.

characterize the properties of adversarial subspaces, but no definitive method yet exists which can reliably discriminate adversarial regions from those in which normal data can be found. Szegedy et al. (2013) argued that adversarial subspaces are low probability regions (not naturally occurring) that are densely scattered in the high dimensional representation space of DNNs. However, a linear formulation argues that adversarial subspaces span a contiguous multidimensional space, rather than being scattered randomly in small pockets (Goodfellow et al., 2014; Warde-Farley et al., 2016). Tanay & Griffin (2016) further emphasize that adversarial subspaces lie close to (but not on) the data submanifold. Similarly, it has also been found that the boundaries of adversarial subspaces are close to legitimate data points in adversarial directions, and that the higher the number of orthogonal adversarial directions of these subspaces, the more transferable they are to other models (Tramèr et al., 2017). To summarize, with respect to the manifold model of data, the known properties of adversarial subspaces are: (1) they are of low probability, (2) they span a contiguous multidimensional space, (3) they lie off (but are close to) the data submanifold, and (4) they have class distributions that differ from that of their closest data submanifold.

Among adversarial defense/detection techniques, *Kernel Density* (KD) estimation has been proposed as a measure to identify adversarial subspaces (Feinman et al., 2017). Carlini & Wagner (2017a) demonstrated the usefulness of KD-based detection, taking advantage of the low probability density generally associated with adversarial subspaces. However, in this paper we will show that kernel density is not effective for the detection of some forms of attack. In addition to kernel density, there are other density-based measures, such as the number of nearest neighbors within a fixed distance, and the mean distance to the $k$ nearest neighbors ($k$-mean distance). Again, these measures have limitations for the characterization of local adversarial regions. For example, in Figure 1 the three density measures fail to differentiate an adversarial example (red star) from a normal example (black cross), as the two examples are locally surrounded by the same number of neighbors (50), and have the same $k$-mean distance (KM=0.19) and kernel density (KD=0.92).

As an alternative to density measures, Figure 1 leads us to consider expansion-based measures of intrinsic dimensionality as a potentially effective method of characterizing adversarial examples. Expansion models of dimensionality assess the local dimensional structure of the data — such models have been successfully employed in a wide range of applications, such as manifold learning, dimension reduction, similarity search and anomaly detection (Amsaleg et al., 2015; Houle, 2017a). Although earlier expansion models characterize intrinsic dimensionality as a property of data sets, the *Local Intrinsic Dimensionality* (LID) fully generalizes this concept to the *local distance distribution* from a reference point to its neighbors (Houle, 2017a;b) — the dimensionality of the local data submanifold in the vicinity of the reference point is revealed by the growth characteristics of the cumulative distribution function. In this paper, we use LID to characterize the intrinsic dimensionality of adversarial regions, and attempt to test how well the estimates of LID can be used to distinguish adversarial examples. Note that the main goal of LID is to characterize properties of adversarial examples, instead of being applied as a pure defense method, which requires stronger assumptions on the current threat model. In Figure 1, the estimated LID of the adversarial example (LID $\approx$ 4.36) is much higher than that of the referenced normal data sample (LID $\approx$ 1.53), illustrating that the estimated LID can efficiently capture the intrinsic dimensional properties of adversarial

regions. In this paper, we aim to study the LID properties of adversarial examples generated using state-of-the-art attack methods. In particular, our contributions are:

- We propose LID for the characterization of adversarial regions of deep networks. We discuss how adversarial perturbation can affect the LID characteristics of an adversarial region, and empirically show that the characteristics of test examples can be estimated effectively using a minibatch of training data.

- Our study reveals that the estimated LID of adversarial examples considered in this paper[1] is significantly higher than that of normal data examples, and that this difference becomes more pronounced in deeper layers of DNNs.

- We empirically demonstrate that the LID characteristics of adversarial examples generated using five state-of-the-art attack methods can be easily discriminated from those of normal examples, and provide a baseline classifier with features based on LID estimates that generally outperforms several existing detection measures on five attacks across three benchmark datasets. Though the adversarial examples considered here are not guaranteed to be the strongest with careful parameter tuning, these preliminary results firmly demonstrate the usefulness of LID measurement.

- We show that the adversarial regions generated by different attacks share similar dimensional properties, in that LID characteristics of a simple attack can potentially be used to detect other more complex attacks. We also show that a naive LID-based detector is robust to the normal low confidence Optimization-based attack of (Carlini & Wagner, 2017a).

## 2    RELATED WORK

In this section, we briefly review the state of the art in both adversarial attack and adversarial defense.

**Adversarial Attack:** A wide range of approaches have been proposed for the crafting of adversarial examples to compromise the performance of DNNs; here, we mention a selection of such works. The Fast Gradient Method (FGM) (Goodfellow et al., 2014) directly perturbs normal input by a small amount along the gradient direction. The Basic Iterative Method (BIM) is an iterative version of FGM (Kurakin et al., 2016). One variant of BIM stops immediately once misclassification has been achieved with respect to the training set (BIM-a), and another iterates a fixed number of steps (BIM-b). For image sets, the Jacobian-based Saliency Map Attack (JSMA) iteratively selects the two most effective pixels to perturb based on the adversarial saliency map, repeating the process until misclassification is achieved (Papernot et al., 2016c). The Optimization-based attack (Opt), arguably the most effective to date, addresses the problem via an optimization framework (Liu et al., 2016; Carlini & Wagner, 2017b).

**Adversarial Defense:** A number of defense techniques have been introduced, including adversarial training (Goodfellow et al., 2014), distillation (Papernot et al., 2016d), gradient masking (Gu & Rigazio, 2014), and feature squeezing (Xu et al., 2017). However, these defenses can generally be evaded by Opt attacks, either wholly or partially (Carlini & Wagner, 2017a; He et al., 2017; Li & Vorobeychik, 2014; 2015). Given the inherent challenges for adversarial defense, recent works have instead focused on detecting adversarial examples. These works attempt to discriminate adversarial examples (positive class) from both normal and noisy examples (negative class), based on features extracted from different layers of a DNN. Detection subnetworks based on activations (Metzen et al., 2017), a cascade detector based on the PCA projection of activations (Li & Li, 2016), an augmented neural network detector based on statistical measures, a learning framework that covers unexplored space in vulnerable models (Rouhani et al., 2017; 2018), a logistic regression detector based on KD, and Bayesian Uncertainty (BU) features (Grosse et al., 2017) are a few such works. However, a recent study by Carlini & Wagner (2017a) has shown that these detection methods can be vulnerable to attack as well.

---

[1]Since our goal is to provide a proof-of-concept for the potential application of LID, we consider only the state-of-the-art methods to generate adversarial examples using default parameters without tuning the parameters to explore the strongest attacks under different conditions.

## 3 LOCAL INTRINSIC DIMENSIONALITY

In the theory of intrinsic dimensionality, classical expansion models (such as the expansion dimension and generalized expansion dimension (Karger & Ruhl, 2002; Houle et al., 2012)) measure the rate of growth in the number of data objects encountered as the distance from the reference sample increases. As an intuitive example, in Euclidean space, the volume of an $m$-dimensional ball grows proportionally to $r^m$, when its size is scaled by a factor of $r$. From this rate of volume growth with distance, the expansion dimension $m$ can be deduced as:

$$\frac{V_2}{V_1} = \left(\frac{r_2}{r_1}\right)^m \Rightarrow m = \frac{\ln(V_2/V_1)}{\ln(r_2/r_1)}. \tag{1}$$

By treating probability mass as a proxy for volume, classical expansion models provide a *local view* of the dimensional structure of the data, as their estimation is restricted to a neighborhood around the sample of interest. Transferring the concept of expansion dimension to the statistical setting of continuous distance distributions leads to the formal definition of LID (Houle, 2017a).

**Definition 1** (Local Intrinsic Dimensionality).
*Given a data sample $x \in X$, let $R > 0$ be a random variable denoting the distance from $x$ to other data samples. If the cumulative distribution function $F(r)$ of $R$ is positive and continuously differentiable at distance $r > 0$, the LID of $x$ at distance $r$ is given by:*

$$\mathrm{LID}_F(r) \triangleq \lim_{\epsilon \to 0} \frac{\ln\big(F((1+\epsilon) \cdot r)/F(r)\big)}{\ln(1+\epsilon)} = \frac{r \cdot F'(r)}{F(r)}, \tag{2}$$

*whenever the limit exists.*

$F(r)$ is analogous to the volume $V$ in Equation (1); however, we note that the underlying distance measure need not be Euclidean. The last equality of Equation (2) follows by applying L'Hôpital's rule to the limits (Houle, 2017a). The local intrinsic dimension at $x$ is in turn defined as the limit, when the radius $r$ tends to zero:

$$\mathrm{LID}_F = \lim_{r \to 0} \mathrm{LID}_F(r). \tag{3}$$

$\mathrm{LID}_F$ describes the relative rate at which its cumulative distance function $F(r)$ increases as the distance $r$ increases from 0, and can be estimated using the distances of $x$ to its $k$ nearest neighbors within the sample (Amsaleg et al., 2015).

In the ideal case where the data in the vicinity of $x$ is distributed uniformly within a submanifold, $\mathrm{LID}_F$ equals the dimension of the submanifold; however, in general these distributions are not ideal, the manifold model of data does not perfectly apply, and $\mathrm{LID}_F$ is not an integer. Nevertheless, the local intrinsic dimensionality does give a rough indication of the dimension of the submanifold containing $x$ that would best fit the data distribution in the vicinity of $x$. We refer readers to Houle (2017a;b) for more details concerning the LID model.

**Estimation of LID:** According to the branch of statistics known as extreme value theory, the smallest $k$ nearest neighbor distances could be regarded as extreme events associated with the lower tail of the underlying distance distribution. Under very reasonable assumptions, the tails of continuous probability distributions converge to the Generalized Pareto Distribution (GPD), a form of power-law distribution (Coles et al., 2001). From this, Amsaleg et al. (2015) developed several estimators of LID to heuristically approximate the true underlying distance distribution by a transformed GPD; among these, the Maximum Likelihood Estimator (MLE) exhibited a useful trade-off between statistical efficiency and complexity. Given a reference sample $x \sim \mathcal{P}$, where $\mathcal{P}$ represents the data distribution, the MLE estimator of the LID at $x$ is defined as follows:

$$\widehat{\mathrm{LID}}(x) = -\left(\frac{1}{k}\sum_{i=1}^{k}\log\frac{r_i(x)}{r_k(x)}\right)^{-1}. \tag{4}$$

Here, $r_i(x)$ denotes the distance between $x$ and its $i$-th nearest neighbor within a sample of points drawn from $\mathcal{P}$, where $r_k(x)$ is the maximum of the neighbor distances. In practice, the sample set is drawn uniformly from the available training data (omitting $x$ itself), which itself is presumed to have been randomly drawn from $\mathcal{P}$. We emphasize that the LID defined in Equation (3) is a *theoretical* quantity, and that $\widehat{\text{LID}}$ as defined in Equation (4) is its *estimate*. In the remainder of this paper, we will refer to Equation (4) to calculate LID estimates.

# 4   CHARACTERIZING ADVERSARIAL REGIONS

Our aim is to gain a better understanding of adversarial regions, and thereby derive potential defenses and provide new directions for more efficient attacks. We begin by providing some motivation with respect to the manifold model of data as to how adversarial perturbation might affect the LID characteristic of adversarial regions. We then show how a detector can potentially be designed using LID estimates to discriminate between adversarial and normal examples.

**LID of Adversarial Subspaces:** Consider a sample $x \in X$ lying within a data submanifold $S$, where $X$ is a randomly sampled dataset from $\mathcal{P}$ consisting only of normal (unperturbed) examples. Adversarial perturbation of $x$ typically results in a new sample $x'$ whose coordinates differ from those of $x$ by very small amounts. Assuming that $x'$ is indeed a successful adversarial perturbation of $x$, the theoretical LID value associated with $x$ is simply the dimension of $S$, whereas the theoretical LID value associated with $x'$ is the dimension of the adversarial subspace within which it resides. Recent work in Amsaleg et al. (2017) shows that the magnitude of the perturbation required to make changes in the expected nearest neighbor ranking tends to zero as the LID and the data sample size tend to infinity.

Since perturbation schemes generally allow the modification of all data coordinates, they exploit the full degrees of freedom afforded by the representational dimension of the data domain. As pointed out by (Goodfellow et al., 2014; Warde-Farley et al., 2016; Tanay & Griffin, 2016), $x'$ is very likely to lie outside $S$ (but very close to $S$ — in a high-dimensional contiguous space). In applications involving high-dimensional data, the representational dimension is typically far larger than the intrinsic dimension of any given data submanifold, which implies that the theoretical LID of $x'$ is far greater than that of $x$.

In practice, however, the values of LID must be estimated from local data samples. This is typically done by applying an appropriate estimator (such as the MLE estimator shown in Equation (4)) to a $k$-nearest neighborhood of the test samples, for some appropriate fixed choice of $k$. Typically, $k$ is chosen large enough for the estimation to stabilize, but not so large that the sample is no longer local to the test sample. If the dimension of $S$ is reasonably low, one can expect the estimation of the LID of $x$ to be reasonably accurate.

For the adversarial subspace, the samples appearing in the neighborhood of $x'$ can be expected to be drawn from more than one manifold. The proximity of $x'$ to $S$ means that the neighborhood is likely to contain neighbors lying in $S$; however, if the neighborhood were composed mostly of samples drawn from $S$, $x'$ would not likely be an adversarial example. Thus, the neighbors of $x'$ taken together are likely to span a subspace of intrinsic dimensionality much higher than any of these submanifolds considered individually, and the LID estimate computed for $x'$ can be expected to reveal this.

**Efficiency through Minibatch Sampling:** Computing neighborhoods with respect to the entirety of the dataset $X$ can be prohibitively expensive, particularly when the (global) intrinsic dimensionality of $X$ is too high to support efficient indexing. For this reason, when $X$ is large, the computational cost can be reduced by estimating the LID of an adversarial example $x'$ from its $k$-nearest neighbor set within a randomly-selected sample (*minibatch*) of the dataset $X$. Since the LID estimation model regards the distances from $x'$ to the members of $X$ as determined by independently-drawn samples from a distribution $\mathcal{P}$, the estimator can also be applied to the distances induced by any random minibatch, as it too would be drawn independently from the same distribution $\mathcal{P}$.

Provided that the minibatch is chosen sufficiently large so as to ensure that the $k$-nearest neighbor sets remain in the vicinity of $x'$, estimates of LID computed for $x'$ within the minibatch would resemble those computed within the full dataset $X$. Conversely, as the size of the minibatch is reduced, the variance of the estimates would increase. However, if the gap between the true LID values of $x$

and $x'$ is sufficiently large, even an extremely small minibatch size and / or small neighborhood size could conceivably produce estimates whose difference is sufficient to reveal the adversarial nature of $x'$. As we shall show in Section 5.2, discrimination between adversarial and non-adversarial examples turns out to be possible even for minibatch sizes as small as 100, and for neighborhood sizes as small as 20.

**Using LID to Characterize Adversarial Examples:** We next describe how LID estimates can serve as features to train a detector to distinguish adversarial examples. Note that here we only aim to train a baseline classifier to demonstrate how well LID can characterize adversarial examples. Robust detection taking different attack variations into account, such as attack confidence, will be left as future work. Our methodology requires that training sets be comprised of three types of examples: adversarial, normal and noisy. This replicates the methodology used in (Feinman et al., 2017; Carlini & Wagner, 2017a), where the rationale for including noisy examples is that DNNs are required to be robust to random input noise (Fawzi et al., 2016) and noisy inputs should not be identified as adversarial attacks. A classifier can be trained by using the training data to construct features for each sample, based on its LID within a minibatch of samples across different layers, where the class label is assigned positive for adversarial examples and assigned negative for normal and noisy examples.

Algorithm 1 describes how the LID features can be extracted for training an LID-based classifier. Given an initial training dataset and a DNN pre-trained on the initial training dataset, the algorithm outputs a classifier trained using LID features. As in previous studies (Carlini & Wagner, 2017a; Feinman et al., 2017), we assume that the initial training dataset is free of adversarial examples — that is, all examples in the dataset are considered 'normal' to begin with. The extraction of LID features first begins with the generation of adversarial and noisy counterparts to normal examples (step 3 and 4) in each minibatch. One minibatch of normal examples ($B_{norm}$) is used for generating 2 counterpart minibatches of examples: one adversarial ($B_{adv}$) and one noisy ($B_{noisy}$). The adversarial examples are generated using an adversarial attack on normal examples (step 3), while noisy examples are generated by adding random noise to normal examples, subject to the constraint that the magnitude of perturbation undergone by a noisy example is the same as the magnitude of perturbation undergone by its counterpart adversarial example (step 4). One minibatch of normal examples is converted to an equal number of adversarial examples after step 3, and an equal number of noisy examples after step 4.

The LID associated with each example (either normal, adversarial or noisy) is estimated from its $k$ nearest neighbors in the *normal* minibatch (steps 12-14), using Equation (4). For any new unknown test example, a minibatch consisting only of normal training examples is used to estimate LID. For each example and each transformation layer in the DNN, an LID estimate is calculated. The distance function needed for this estimate uses the activation values of the neurons in the given layer as inputs (step 7). As will be discussed in Section 5.2, we use all transformation layers, including conv2d, max-pooling, dropout, ReLU and softmax, since we expect adversarial regions to exist in each layer of the DNN representation space. The LID estimates associated with the example are then used as feature values (one feature for each transformation layer). Finally, a classifier (such as logistic regression) is trained using the LID features. Test examples can then be classified by the LID-based classifier to either the positive (adversarial) or negative (non-adversarial) class by means of its LID-based feature values.

# 5 EVALUATING LID-BASED CHARACTERIZATION OF ADVERSARIAL EXAMPLES

In this section, we evaluate the discrimination power of LID-based characterization against five adversarial attack strategies — FGM, BIM-a, BIM-b, JSMA, and Opt, as introduced in Section 2. These attack strategies were selected for our experiments due to their reported effectiveness and their diversity. For each of the 5 forms of attack, the LID detector is compared with the state-of-the-art detection measures KD and BU as discussed in Section 2, with respect to three benchmark image datasets: MNIST (LeCun et al., 1990), CIFAR-10 (Krizhevsky & Hinton, 2009) and SVHN (Netzer et al., 2011). Each of these three datasets is associated with a designated training set and test set. Before reporting and discussing the results, we first describe the experimental setup.

---

**Algorithm 1** Training phase for LID-based adversarial classifier

---

**Input:**
    $X$: a dataset of normal examples
    $H(x)$: a pre-trained DNN with $L$ transformation layers
    $k$: the number of nearest neighbors for LID estimation
**Output:**
    Detector(LID)                                                ▷ a detector
  1: LID$_{neg}$=[], LID$_{pos}$=[]
  2: **for** $B_{norm}$ in $X$ **do**                       ▷ $B_{norm}$: a minibatch of normal examples
  3:     $B_{adv}$ := adversarial attack $B_{norm}$        ▷ $B_{adv}$: a minibatch of adversarial examples
  4:     $B_{noisy}$ := add random noise to $B_{norm}$        ▷ $B_{noisy}$: a minibatch of noisy examples
  5:     $N = |B_{norm}|$                                ▷ number of examples in $B_{norm}$
  6:     LID$_{norm}$, LID$_{noisy}$, LID$_{noisy}$ = zeros$[N, L]$
  7:     **for** $i$ in $[1, L]$ **do**
  8:         $A_{norm} = H^i(B_{norm})$                 ▷ $i$-th layer activations of $B_{norm}$
  9:         $A_{adv} = H^i(B_{adv})$                   ▷ $i$-th layer activations of $B_{adv}$
10:         $A_{noisy} = H^i(B_{noisy})$              ▷ $i$-th layer activations of $B_{noisy}$
11:         **for** $j$ in $[1, N]$ **do**
12:             LID$_{norm}[j, i] = -\left(\frac{1}{k} \sum_{i=1}^{k} \log \frac{r_i(A_{norm}[j], A_{norm})}{r_k(A_{norm}[j], A_{norm})}\right)^{-1}$
13:             LID$_{adv}[j, i] = -\left(\frac{1}{k} \sum_{i=1}^{k} \log \frac{r_i(A_{adv}[j], A_{norm})}{r_k(A_{adv}[j], A_{norm})}\right)^{-1}$
14:             LID$_{noisy}[j, i] = -\left(\frac{1}{k} \sum_{i=1}^{k} \log \frac{r_i(A_{noisy}[j], A_{norm})}{r_k(A_{noisy}[j], A_{norm})}\right)^{-1}$
15:              ▷ $r_i(A[j], A_{norm})$: the $L_2$ distance of $A_-[j]$ to its $i$-th nearest neighbor in $A_{norm}$
16:         **end for**
17:     **end for**
18:     LID$_{neg}$.append(LID$_{norm}$), LID$_{neg}$.append(LID$_{noisy}$)
19:     LID$_{pos}$.append(LID$_{adv}$)
20: **end for**
21: Detector(LID) = train a classifier on (LID$_{neg}$, LID$_{pos}$)

---

## 5.1 EXPERIMENTAL SETUP

**Training and Testing:** For each of the three image datasets, a DNN classifier was independently pretrained on its designated training set (the *pre-train* set), and its designated test set was used for testing (the *pre-test* set). Any *pre-test* images not correctly classified were discarded, and the remaining images were subdivided into *train* (80%) and *test* (20%) sets for subsequent processing. Both of these sets were randomly partitioned into minibatches of size 100, for later use in the computation of LID characteristics.

The LID-, KD- and BU-based detectors were trained separately on the *train* set using the scheme in Algorithm 1, with the calculation of LID estimates replaced by KD and BU calculation for their respective detectors. All three detectors were then evaluated against equal numbers of normal, noisy and adversarial images crafted from members of the *test* set, as described in Steps 2-4 of Algorithm 1. The LID, KD and BU characteristics of those test images were then generated as shown in Steps 1-19 of Algorithm 1. It should be noted that no images of the *test* set were examined during any of the training processes, so as to avoid cross contamination. The adversarial examples for both training and testing were generated by applying one of the five selected attacks. Following the procedure outlined in Feinman et al. (2017), the noisy examples for the JSMA attack were crafted by changing the values of a randomly-selected set of pixels to either their minimum or maximum (determined randomly), where the number of pixels to be adjusted was chosen to be equal to the number of pixels perturbed in the generation of adversarial examples. For the other attack strategies, $L_2$ Gaussian noise was added to the pixel values instead of setting them to their minimum or maximum. As suggested by Feinman et al. (2017); Carlini & Wagner (2017a), we used the logistic regression classifier as detector, and report its AUC score as the metric for performance.

**Deep Neural Networks for Pretraining:** The pretrained DNN used for MNIST was a 5-layer ConvNet with max-pooling and dropout. It achieved 99.29% classification accuracy on (normal)

*pre-test* images. For CIFAR-10, a 12-layer ConvNet with max-pooling and dropout was used. This model reported an accuracy of 84.56% on (normal) *pre-test* images. For SVHN, we trained a 6-layer ConvNet with max-pooling and dropout. It achieved 92.18% accuracy on (normal) *pre-test* images. We deliberately did not tune the DNNs, as their performance was close to the state-of-the-art and could thus be considered sufficient for use in an adversarial study (Feinman et al., 2017).

**Parameter Tuning:** We tuned the bandwidth ($\sigma$) parameter for KD, and the number of nearest neighbors ($k$) for LID, using nested cross validation within the training set (*train*). Using the AUC values of detection performance, the bandwidth was tuned using a grid search over the range $[0, 10)$ in log-space, and neighborhood size was tuned using a grid search over the range $[10, 100)$ with respect to a minibatch of size 100. For a given dataset, the parameter setting selected was the one with highest AUC averaged across all attacks. The optimal bandwidths chosen for MNIST, CIFAR-10 and SVHN were 3.79, 0.26, and 1.0, respectively, while the value of $k$ for LID estimation was set to 20 for MNIST and CIFAR-10, and 30 for SVHN. For BU, we chose the number of prediction runs to be $T = 50$ in all experiments. We did not tune this parameter, as it is not considered to be sensitive for choices of $T$ greater than 20 (Carlini & Wagner, 2017a).

Our implementation is based on the detection framework of Feinman et al. (2017). For FGM, JSMA, BIM-a, and BIM-b attack strategies, we used the *cleverhans* library (Papernot et al., 2016a), and for the Opt attack strategy, we used the author's implementation (Carlini & Wagner, 2017b). We scaled all image feature values to the interval $[0, 1]$. Our code is available for download at `https://github.com/xingjunm/lid_adversarial_subspace_detection`.

## 5.2 LID Characteristics of Adversarial Examples

We provide empirical results showing the LID characteristics of adversarial examples generated by Opt, the most effective of the known attack strategies. The left subfigure in Figure 2 shows the LID scores (at the softmax layer) of 100 randomly selected normal, noisy and adversarial (Opt) examples from the CIFAR-10 dataset. We observe that at this layer, the LID scores of adversarial examples are significantly higher than those of normal or noisy examples. This supports our expectation that adversarial regions have higher intrinsic dimensionality than normal data regions (as discussed in Section 4). It also suggests that the transition from normal example to adversarial example may follow directions in which the complexity of the local data submanifold significantly increases, leading to an increase in estimated LID values.

In the right subfigure of Figure 2, we further show that the LID scores of adversarial examples are more easily discriminated from those of other examples at deeper layers of the network. The 12-layer ConvNet used for CIFAR-10 consists of 26 transformation layers: the input layer ($L_0$), conv2d/max-pooling ($L_{1-17}$), dense/dropout ($L_{18-24}$) and the final softmax layer ($L_{25}$). The estimated LID characteristics of adversarial examples become distinguishable (detection AUC$> 0.5$) at the dense layers ($L_{18-24}$), and significantly different at the softmax layer ($L_{25}$). This suggests that the fully-connected and softmax transformations may be more sensitive to adversarial perturbations than convolutional transformations. Plots of LID scores for the MNIST and SVHN datasets can be found in Appendix A.2.

With regard to the stability of performance based on parameter variation ($k$ for LID, or bandwidth for KD), we can see from Figure 3 that LID is more stable than KD, exhibiting less variation in AUC as the parameter varies. From this figure, we also see that KD requires significantly different optimal settings for different types of data. For simpler datasets such as MNIST and SVHN, KD requires quite high bandwidth choices for best performance.

## 5.3 Analysis of LID Properties

**LID Outperforms KD and BU:** We compare the performance of LID-based detection with that of detectors trained with features of KD and BU individually, as well as a detector trained with a combination of KD and BU features (denoted as 'KD+BU'). As shown in Table 1, LID outperforms the KD and BU measures (both individually and combined) by large margins on all attack strategies tested, across all datasets tested. For the most effective attack strategy known to date, the Opt attack, the LID-based detector achieved AUC scores of 99.24%, 98.94% and 97.60% on MNIST, CIFAR-10 and SVHN respectively, compared to AUC scores of 95.35%, 93.77% and 90.66% for

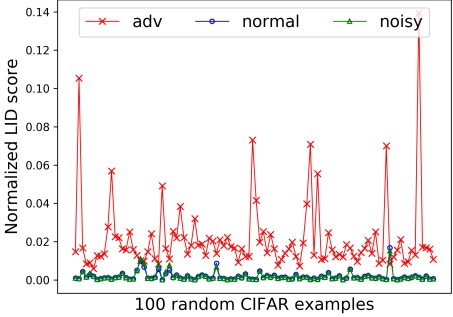 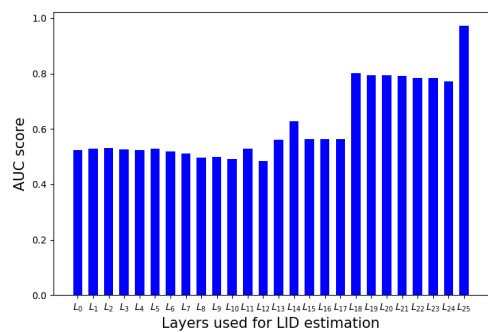

Figure 2: The left-hand figure shows the LID scores (at the softmax layer) of 100 normal (blue), noisy (green), and Opt attack (red x-cross) examples from the CIFAR-10 dataset. The scores have been scaled to the interval [0,1] using min-max normalization. The blue and green lines appear superimposed due to similarities in the LID scores for normal and noisy examples. The right-hand figure shows the detection performance (AUC) based on LID scores computed at different layers. $L_i$ denotes the $i$-th transformation layer.

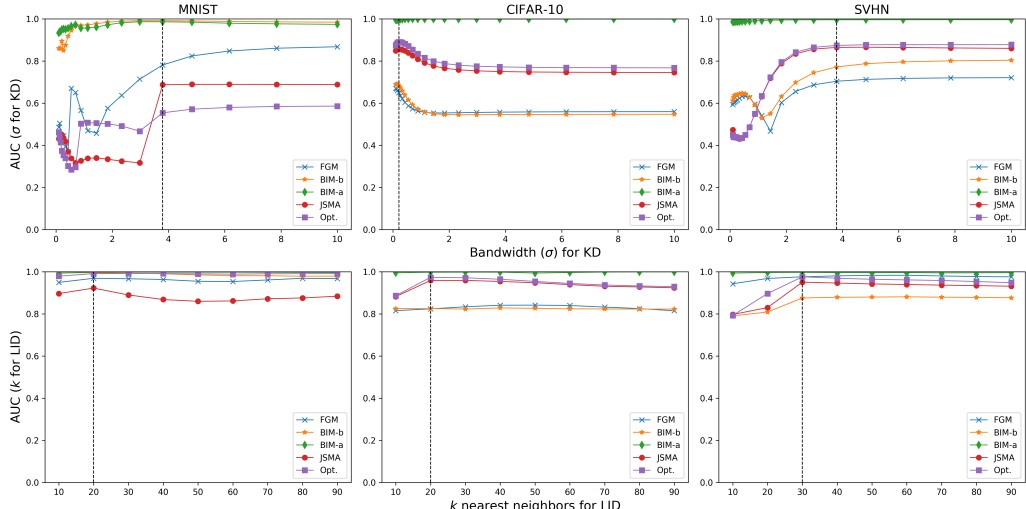

Figure 3: Top row: tuning bandwidth $\sigma$ for KD using a grid search over the range $[0, 10)$ in log-space, separately for each dataset. Bottom row: tuning $k$ for LID using a grid search over the range $[10, 100)$ for minibatch size 100, separately for each dataset. The vertical dashed lines denote the selected parameter choice.

the detector based on KD and BU. This strong performance suggests that LID is a highly promising characteristic for the discrimination of adversarial examples and regions. We also note that KD was not effective for the FGM, JSMA and BIM-a attack strategies, whereas the BU measure failed to detect most FGM and BIM-b attacks on the MNIST dataset.

**Generalizability Analysis:** It is natural to consider the question of whether samples of one attack strategy may be detected by a model that has been trained on samples of a different attack strategy. We conduct a preliminary investigation of this issue by studying the generalizability of KD, BU and LID for detecting previously unseen attack strategies on the CIFAR-10 dataset. The KD, BU and LID detectors are trained on samples of the simplest attack strategy, FGM, and then tested on samples of the more complex attacks BIM-a, BIM-b, JSMA and Opt. The training and test datasets are generated in the same way as in our previous experiments with only the FGM attack applied on the *train* set while the other attacks applied separately on the *test* set. The test attack data is standardized by scaling so as to fit the training data. The results are shown in Table 2, from which

Table 1: A comparison of the discrimination power (AUC score (%) of a logistic regression classifier) among LID, KD, BU, and KD+BU. The AUC score is computed for each attack strategy on each dataset, and the best results are highlighted in **bold**.

| Dataset | Feature | FGM | BIM-a | BIM-b | JSMA | Opt |
|---|---|---|---|---|---|---|
| MNIST | KD | 78.12 | 98.14 | 98.61 | 68.77 | 95.15 |
| | BU | 32.37 | 91.55 | 25.46 | 88.74 | 71.30 |
| | KD+BU | 82.43 | 99.20 | 98.81 | 90.12 | 95.35 |
| | LID | **96.89** | **99.60** | **99.83** | **92.24** | **99.24** |
| CIFAR-10 | KD | 64.92 | 68.38 | 98.70 | 85.77 | 91.35 |
| | BU | 70.53 | 81.60 | 97.32 | 87.36 | 91.39 |
| | KD+BU | 70.40 | 81.33 | 98.90 | 88.91 | 93.77 |
| | LID | **82.38** | **82.51** | **99.78** | **95.87** | **98.94** |
| SVHN | KD | 70.39 | 77.18 | 99.57 | 86.46 | 87.41 |
| | BU | 86.78 | 84.07 | 86.93 | 91.33 | 87.13 |
| | KD+BU | 86.86 | 83.63 | 99.52 | 93.19 | 90.66 |
| | LID | **97.61** | **87.55** | **99.72** | **95.07** | **97.60** |

we see that the LID detector trained on FGM can accurately detect the much more complex attacks of the other strategies. The KD and BU characteristics can also achieve good performance on this transfer learning task, but are less consistent than our proposed LID characteristic. The results appear to indicate that the adversarial regions generated by different attack strategies possess similar dimensional properties.

It is worth mentioning that the BU detector trained on the FGM attack generalizes poorly to detect BIM-b adversarial examples (AUC=2.65%). This may due to the fact that BIM-b performs a fixed number of perturbations (50 in our setting) that likely extend well beyond the classification boundary. Such perturbed adversarial examples tend to possess Bayesian model uncertainties even lower than normal examples under dropout randomization, as dropping out a certain proportion of their representations (50% in our setting) would not lead to high prediction variance. This is consistent with the results reported in Feinman et al. (2017): only 4% of BIM-b adversarial examples, in contrast to at least 74.7% of adversarial examples of other attack strategies, exhibit higher Bayesian uncertainties than normal examples. It is particularly interesting to see that detectors trained on the FGM attack strategy can sometimes achieve better performance when used to identify the other attacks. An extensive study of detection generalizability across all attack strategies is an interesting topic for future work.

Table 2: This table of AUC scores (%) shows the generalizability of detectors trained on the FGM attack strategy (row) to other forms of attack (column), with respect to the CIFAR-10 dataset. The best results are indicated in **bold** font.

| Train \ Test | | FGM | BIM-a | BIM-b | JSMA | Opt |
|---|---|---|---|---|---|---|
| FGM | KD | 64.92 | 69.15 | 89.71 | 85.72 | 91.22 |
| | BU | 70.53 | 81.67 | 2.65 | 86.79 | 91.27 |
| | LID | **82.38** | **82.30** | **91.61** | **89.93** | **93.32** |

**Effect of Larger Minibatch Sizes in LID Estimation:** In the estimation of LID values, a default minibatch size of 100 was used, with a view to ensuring efficiency. Even though experimental analysis has shown that the MLE estimator of LID is not stable on such small samples (Amsaleg et al., 2015), this is more than adequately compensated for by the learning process in LID-based detection, as evidenced by the superior performance shown in Table 1. However, it is an interesting question as to whether the use of larger minibatch sizes could further improve the performance (as measured by AUC) without incurring unreasonably high computational cost. Figure 5 in Appendix A.3 illustrates the effect of using a minibatch size of 1000 for different choices of $k$. It does indicate that increasing the batch size can improve the detection performance even further. A comprehensive investigation of the tradeoffs among minibatch size, LID estimation accuracy, and detection performance is an interesting direction for future work.

Table 3: The failure rate (%) of an adaptive attack targeting the LID-based detector.

|  | MNIST | CIFAR-10 | SVHN |
|---|---|---|---|
| Scenario 1 (LID at all layers): Attack Failure Rate | 100 | 100 | 100 |
| Scenario 2 (LID at one layer): Attack Failure Rate | 100 | 95.7 | 97.2 |

**Adaptive Attack Against LID Measurement:** To further evaluate the robustness of our LID-based detector, we applied an adaptive Opt attack in a white-box setting. Similar to the strategy used in Carlini & Wagner (2017a) to attack the KD-based detector, we used an Opt $L_2$ attack with a modified adversarial objective:

$$\text{minimize } \|x - x_{adv}\|_2^2 + \alpha \cdot \big(\ell(x_{adv}) + \ell(\text{LID}(x_{adv}))\big) \tag{5}$$

where $\alpha$ is a constant balancing between the amount of perturbation and the adversarial strength, and the LID scores are computed at the pre-softmax layer.

We test two different scenarios for detection. In the first scenario, we use LID features as described in Algorithm 1. In the second scenario, we use LID scores only at the pre-softmax layer. Since the Opt attack uses only the pre-softmax activation output to guide the perturbation, the latter scenario allows a fair comparison to be made (Carlini & Wagner, 2017b;a). The optimal constant $\alpha$ is determined via an internal binary search for $\alpha \in [10^{-3}, 10^6]$. The rationale for the minimization of the LID characteristic in Equation (5) is that adversarial examples have higher LID characteristics than normal examples, as we have demonstrated in Section 5.2.

We applied the adaptive attack on 1000 normal images randomly chosen from the detection test set (*test*). The deep networks used were the same ConvNet configurations as used in our previous experiments. To evaluate attack performance, instead of AUC as measured in the previous sections, we report accuracy as suggested by Carlini & Wagner (2017a). We see from Table 3 that the adaptive attack in Scenario 2 fails to find any valid adversarial example $100\%$, $95.7\%$ and $97.2\%$ of the time on MNIST, CIFAR-10 and SVHN respectively. In addition, when trained on all transformation layers (Scenario 1), the LID-based detector still correctly detected the attacks $100\%$ of the time. Based on these results, we can conclude that integrating LID into the adversarial objective (increasing the complexity of the attack) does not make detection more difficult for our method. This is in contrast to the work of Carlini & Wagner (2017a), who showed that incorporating kernel density into the objective function makes detection substantially more difficult for the KD method.

## 6 DISCUSSION AND CONCLUSION

In this paper, we have addressed the challenge of understanding the properties of adversarial regions, particularly with a view to detecting adversarial examples. We characterized the dimensional properties of adversarial regions via the use of Local Intrinsic Dimensionality (LID), and showed how these could be used as features in an adversarial example detection process. Our empirical results suggest that LID is a highly promising measure for the characterization of adversarial examples, one that can be used to deliver state-of-the-art discrimination performance. From a theoretical perspective, we have provided an initial intuition as to how LID is an effective method for characterizing adversarial attack, one which complements the recent theoretical analysis showing how increases in LID effectively diminish the amount of perturbation required to move a normal example into an adversarial region (with respect to 1-NN classification) (Amsaleg et al., 2017). Further investigation in this direction may lead to new techniques for both adversarial attack and defense.

In the learning process, the activation values at each layer of the LID-based detector can be regarded as a transformation of the input to a space in which the LID values have themselves been transformed. A full understanding of LID characteristics should take into account the effect of DNN transformations on these characteristics. This is a challenging question, since it requires a better understanding of the DNN learning processes themselves. One possible avenue for future research may be to model the dimensional characteristics of the DNN itself, and to empirically verify how they influence the robustness of DNNs to adversarial attacks.

Another open issue for future research is the empirical investigation of the effect of LID estimation quality on the performance of adversarial detection. As evidenced by the improvement in perfor-

mance observed when increasing the minibatch size from 100 to 1000 (Figure 5 in Appendix A.3), it stands to reason that improvements in estimator quality or sampling strategies could both be beneficial in practice.

ACKNOWLEDGMENTS

James Bailey is in part supported by the Australian Research Council via grant number DP170102472. Michael E. Houle is in part supported by JSPS Kakenhi Kiban (B) Research Grant 15H02753. Bo Li and Dawn Song are partially supported by Berkeley Deep Drive, the Center for Long-Term Cybersecurity, and FORCES (Foundations Of Resilient CybEr-Physical Systems), which receives support from the National Science Foundation (NSF award numbers CNS-1238959, CNS-1238962, CNS-1239054, CNS-1239166).

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

## A APPENDIX

### A.1 STATISTICS OF ADVERSARIAL ATTACK STRATEGIES

Table 4: The $L_2$ mean perturbation and model accuracy (%) on adversarial examples.

|  | MNIST | | CIFAR | | SVHN | |
|---|---|---|---|---|---|---|
|  | $L_2$ | Acc. | $L_2$ | Acc. | $L_2$ | Acc. |
| FGM | 6.26 | 11.09 | 2.74 | 3.15 | 7.09 | 6.17 |
| BIM-a | 2.30 | 10.43 | 0.48 | 0.00 | 0.83 | 0.13 |
| BIM-b | 5.42 | 10.42 | 3.39 | 0.00 | 5.53 | 0.13 |
| JSMA | 5.40 | 10.00 | 3.64 | 0.04 | 3.09 | 0.16 |
| Opt | 4.21 | 3.92 | 0.37 | 0.01 | 0.59 | 0.26 |

### A.2 LID CHARACTERISTICS OF ADVERSARIAL EXAMPLES

Figure 4 illustrates LID characteristics of the most effective attack strategy known to date, Opt, on the MNIST and SVHN datasets. On both datasets, the LID scores of adversarial examples are significantly higher than those of normal or noisy examples. In the right-hand plot, the LID scores of normal examples and its noisy counterparts appear superimposed due to their similarities.

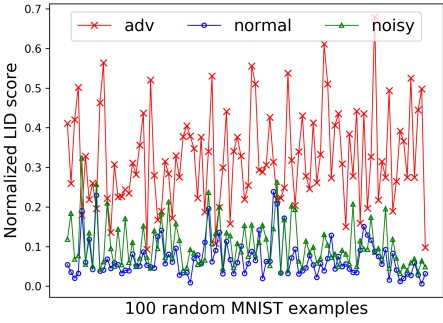 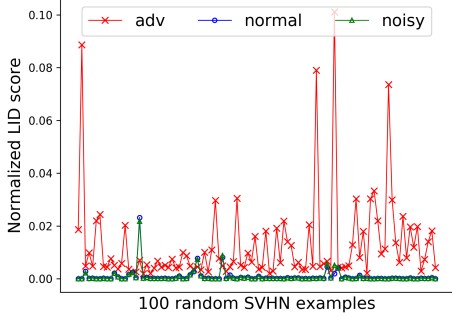

Figure 4: The plots show the normalized LID scores of 100 randomly selected normal (blue), noisy (green) and Opt attack (red x-cross) examples. The noisy and adversarial examples were generated from the normal examples. The left-hand plot shows the scores (at the pre-softmax layer) of MNIST examples, while the right-hand plot shows LID scores (at the softmax layer) of SVHN examples. Normal and noisy example curves appear superimposed in the right-hand figure due to the similarity of their values.

### A.3 Effect of Larger Minibatch Sizes in LID Estimation

Figure 5 shows the discrimination power (detection AUC) of LID characteristics estimated using two different minibatch sizes: the default setting of 100, and a larger size of 1000. The horizontal axis represents different choices of the neighborhood size $k$, from $10\%$ to $90\%$ percent to the batch size. We note that the peak AUC is higher for the larger minibatch size.

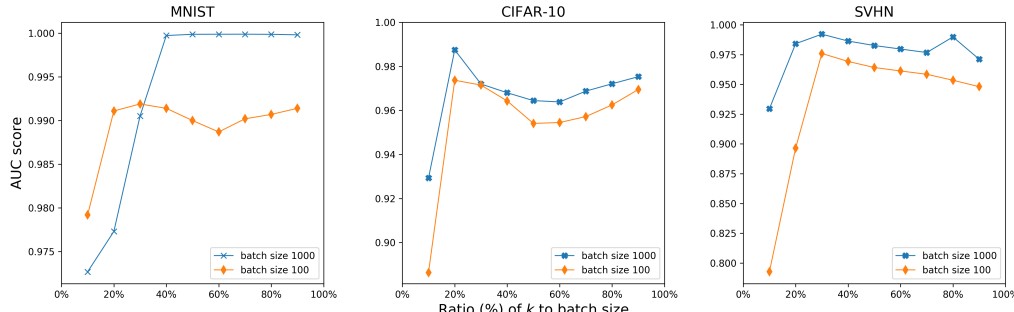

Figure 5: The detection AUC score of LID estimated using different neighborhood sizes $k$ with a larger minibatch size of 1000. The results are shown for the detection of Opt attacks on the MNIST, CIFAR-10 and SVHN datasets.

