# OpenReview forum: "Characterizing Adversarial Subspaces Using Local Intrinsic Dimensionality"
_ICLR.cc/2018/Conference — Accept (Oral)_

### Official Review · AnonReviewer2 · 2017-11-25
**Intrinsic dimensionality around the adversarial example is very different from the one of normal or noisy data**

**Rating:** 8
**Confidence:** 3

**Review:**

The paper considers a problem of adversarial examples applied to the deep neural networks. The authors conjecture that the intrinsic dimensionality of the local neighbourhood of adversarial examples significantly differs from the one of normal (or noisy) examples. More precisely, the adversarial examples are expected to have intrinsic dimensionality much higher than the normal points (see Section 4).  Based on this observation they propose to use the intrinsic dimensionality as a way to separate adversarial examples from the normal (and noisy) ones during the test time. In other words, the paper proposes a particular approach for the adversarial defence.

It turns out that there is a well-studied concept in the literature capturing the desired intrinsic dimensionality: it is called the local intrinsic dimensionality (LID, Definition 1) . Moreover, there is a known empirical estimator of LID, based on the k-nearest neighbours. The authors propose to use this estimator in computing the intrinsic dimensionalities for the test time examples. For every test-time example X the resulting Algorithm 1 computes LID estimates of X activations computed for all intermediate layer of DNN. These values are finally used as features in classifying adversarial examples from normal and noisy ones.

The authors empirically evaluate the proposed technique across multiple state-of-the art adversarial attacks, 3 datasets (MNIST, CIFAR10, and SVHN) and compare their novel adversarial detection technique to 2 other ones recently reported in the literature. The experiments support the conjecture mentioned above and show that the proposed technique *significantly* improves the detection accuracy compared to 2 other methods across all attacks and datasets (see Table 1).

Interestingly, the authors also test whether adversarial attacks can bypass LID-based detection methods by incorporating LID in their design. Preliminary results show that even in this case the proposed method manages to detect adversarial examples most of the time. In other words, the proposed technique is rather stable and can not be easily exploited.

I really enjoyed reading this paper. All the statements are very clear, the structure is transparent and easy to follow. The writing is excellent. I found only one typo (page 8, "We also NOTE that..."), otherwise I don't actually have any comments on the text.

Unfortunately, I am not an expert in the particular field of adversarial examples, and can not properly assess the conceptual novelty of the proposed method. However, it seems that it is indeed novel and given rather convincing empirical justifications, I would recommend to accept the paper.

---

> ### Author Response · Authors · 2017-12-16
> **Typo fixed.**
>
> We are glad that you like our work and would like to thank you for the summary. The typo has been fixed in the updated version.

---

### Official Review · AnonReviewer3 · 2017-11-27
**Not familiar with this research topic.**

**Rating:** 6
**Confidence:** 1

**Review:**

This paper tried to analyze the subspaces of the adversarial examples neighborhood. More specifically, the authors used Local Intrinsic Dimensionality to analyze the intrinsic dimensional property of the subspaces. The characteristics and theoretical analysis of the proposed method are discussed and explained. This paper helps others to better understand the vulnerabilities of DNNs.

---

> ### Author Response · Authors · 2017-12-16
> **Thanks for your reply.**
>
> We appreciate your candor about this research topic. At a high level, although deep neural networks have demonstrated superior performance for many tasks, certain properties which can affect their behavior (such as subspaces, manifold properties) are still not well understood.   A better understanding of these properties can motivate more robust/efficient/effective deep learning models, which can in turn lead to further improving their performance. Adversarial vulnerability is one such property that jeopardizes the reliability of deep neural network learning models, as very small changes on inputs can sometimes lead to completely incorrect predictions (such changed inputs are called adversarial inputs). In this paper, we investigate the expansion dimensional property of the subspaces surrounding such adversarial inputs and show that it can be used as an effective characteristic for detecting such inputs. We hope our work can provide some new insights into adversarial subspaces and their detection.

---

### Official Review · AnonReviewer1 · 2017-11-27
**Intuitive solution to important problem; well written**

**Rating:** 7
**Confidence:** 4

**Review:**

The authors clearly describe the problem being addressed in the manuscript and motivate their solution very clearly. The proposed solution seems very intuitive and the empirical evaluations demonstrates its utility. My main concern is the underlying assumption (if I understand correctly) that the adversarial attack technique that the detector has to handle needs to be available at the training time of the detector. Especially since the empirical evaluations are designed in such a way where the training and test data for the detector are perturbed with the same attack technique. However, this does not invalidate the contributions of this manuscript.

Specific comments/questions:
- (Minor) Page 3, Eq 1: I think the expansion dimension cares more about the probability mass in the volume rather than the volume itself even in the Euclidean setting.
- Section 4: The different pieces of the problem (estimation, intuition for adversarial subspaces, efficiency) are very well described.
- Alg 1, L3: Is this where the normal exmaples are converted to adversarial examples using some attack technique?
- Alg 1, L12: Is LID_norm computed using a leave-one-out estimate? Otherwise, r_1(.) for each point is 0, leading to a somewhat "under-estimate" of the true LID of the normal points in the training set. I understand that it is not an issue in the test set.
- Section 4 and Alg 1: S we do not really care about the "labels/targets" of the examples. All examples in the dataset are considered "normal" to start with. Is this assuming that the "initial training set" which is used to obtain the "pre-trained DNN" free of adversarial examples?
- Section 5, Experimental Setup: Seems like normal points in the test set would get lesser values if we are not doing the "leave-one-out" version of the estimation.
- Section 5: The authors have done a great job at evaluating every aspect of the proposed method.

---

> ### Author Response · Authors · 2017-12-16
> **Comments have been addressed in updated paper.**
>
> Thank you very much for these comments. We address them in detail below.
> Q1: The adversarial attack technique needs to be available for training.
> A1: Thank you for highlighting this. The ability to detect unseen adversarial attacks is an interesting issue.   We have conducted some additional experiments to evaluate the generalizability of our LID based detector, see “Generalizability Analysis”, Section 5.3. The result illustrates that our LID-based detector generalizes well to detect previously unseen adversarial attacks.
>
> Q2: (Minor) Page 3, Eq 1: The expansion dimension cares more about the probability mass.
> A2: Yes, we agree. The suggested explanation has been added to Paragraph 1, Section 3.
>
> Q3: Alg 1, L3: is this where the adversarial attacks are applied?
> A3: Yes. We have updated the description of the algorithm to clarify this (see Paragraph 2 in "Using LID to Characterize Adversarial Examples", Section 4).
>
> Q4: Alg 1, L12 & Section 5, Experimental Setup: leave-one-out estimate?
> A4: Yes, the query point x is "left out". We have added extra explanations of how Eq (4) (as used in L12-14, Alg 1) works in the last paragraph of Section 3.
>
> Q5: Section 4 and Alg 1: assuming training data is free of adversarial examples?
> A5: Yes. This is a reasonable assumption and is the one which has been made in previous work. We have highlighted this in the 2nd paragraph of "Using LID to Characterize Adversarial Examples", Section 4.

---

> > ### Public Comment · (anonymous) · 2017-12-19
> > **BIM-b in Table 2?**
> >
> > Table 2: What happens in the case of BIM-b?
> >
> > Also, it seems very strange that for many cases in Table 2, training with FGM seems to give better detection rates that training with the actual attack used? (In 7/15 cases).
> >
> > Also, the 'opt' approach seems to fail on KD as well, while it has been shown to break KD easily earlier in literature (as mentioned in intro). Why does 'opt' fail on KD in the current setting? Was something changed with KD?
> >
> > Section 5.2 of Carlini & Wagner 2017a indicates an approach to make these attacks successful in the MNIST setting. And, they suggest that KD completely breaks down in the CIFAR-10 setting whilst you report a 91% accuracy. I suspect there might be some issue with implementing the CW attack, since the numbers are in complete contrast to that reported in 2017a, to be sure I would just check with the implementation at https://github.com/carlini/nn_breaking_detection/blob/master/density_estimation.py.

---

> > > ### Author Response · Authors · 2017-12-20
> > > **Re: BIM-b in Table 2?**
> > >
> > > Thank you for these comments. We have uploaded a new version to address them.
> > >
> > > -- Understanding of Table 2:
> > > We would like to clarify that the detectors used in Table 2 to detect other attacks, in the previous version, were trained with 20% more data than those used in Table 1 --- that is, the detectors in Table 2 were trained on the training set (80%) plus the test set (20%), whereas those in Table 1 were trained only on the training set (80%). In the latest version of the paper, we have fixed this inconsistency and provided updated results for Table 2, using exactly the same amount of training data (80%) as was used for Table 1. Moreover, we have also provided additional explanations about the exceptionally poor performance of BU measure transferring from FGM to BIM-b.
> > >
> > > Meanwhile, we would like to point out that the Opt (or CW) attack we used in this paper is its general L2 version, different to many of its variants designed to attack specific defenses. Code on GitHub by the authors of Opt: https://github.com/carlini/nn_robust_attacks/blob/master/l2_attack.py
> > > The reason is that we are more interested in the understanding of the shared properties across different types of attack strategies including Opt and also many others, so that to motivate defense against adversarial attacks in general (not limited to Opt attack). You are very welcome to check the consistency of the code --- we appreciate your interest in this.
> > >
> > > The following responses are related to two papers: 1) the original defense paper of KD and BU (Feinman et al. (2017)), and 2) the latest attack paper (Carlini&Wagner (2017a)).
> > >
> > > -- KD works on CIFAR-10?
> > > Yes, our result in Table 1 indicates this. This contradicts the statement in Paper 2 Sect. 5.2, where it says that KD cannot work on CIFAR, as 80% of the time the adversarial example has a higher likelihood score than the original image. However, our result is consistent with the result in the original Feinman paper (Table 2, Paper 1). We found not only that Opt plus KD can detect Opt, but also that the simpler attack FGM plus KD can detect Opt (Table 2). The deeper reasons behind this require further exploration.
> > >
> > > --‘Opt’ seems to fail against the KD detector?
> > > Yes. But it is the general Opt L2 attack that failed the KD detector, not the KD-adapted version of Opt. In Paper 2, the KD detector has been attacked by a variant of Opt specifically adapted to target KD measure. But we did not use this version of Opt, as we are more interested in the general Opt attack rather than a specially-adapted version of it.

---

> > > > ### Public Comment · ~Nicholas_Carlini1 · 2017-12-20
> > > > **Re: BIM-b in Table 2?**
> > > >
> > > > I agree with the comment that it looks like there is something interesting going on here (it's not immediately clear why training on FGSM will make it do better on optimization methods). However: it does look like the authors properly evaluate the defense given equation (5).
> > > >
> > > > Performing the adaptive attack is what is missing from most prior work (and, indeed, even from many of the papers submitted here). I would much rather see a paper with a proper evaluation and some followup questions on the results than one that omits the evaluation entirely and therefore has no questions. I hope the authors are not penalized for this.

---

> > > > > ### Public Comment · (anonymous) · 2017-12-21
> > > > > **What happens if the adversary has knowledge of the logistic regression model?**
> > > > >
> > > > > Do you have insights on what might happen if the adversary actually has full knowledge of the logistic regression model trained to distinguish based on the LID score (or he/she could train one himself/herself, too)?

---

> > > > > > ### Author Response · Authors · 2017-12-22
> > > > > > **white-box attacking LID detector**
> > > > > >
> > > > > > Good question. In Section "Robustness to Adaptive Attack", we have shown that simply attacking the LID score (towards decreasing the LID scores of adversarial examples) is not effective. Attacking the logistic regression model itself can be expected to have similar results if directly integrate the detection model into the adversarial objective as we did for the LID score. In this paper, we show that adversarial examples tend to transit from a low dimensional submanifold to a more "complex" submanifold. A more interesting question is to what extent an attack strategy relies on such transition to find a valid solution. Our proposed LID is not a perfect solution for adversarial detection, after all, it is not 100% accurate in the detection of adversarial examples. In the future, we will investigate more forms of adapted Opt attacks against our LID detector and develop a more in-depth understanding of the detected/escaped adversarial examples. We would very much like to see Nicholas's response to this.

---

> > > > > > > ### Public Comment · (anonymous) · 2017-12-23
> > > > > > > **Great stuff!**
> > > > > > >
> > > > > > > Thanks for your responses. They do shed some light on the results. More questions out of curiosity: What is the recall rate? That is, what fraction of test/train examples are detected as adversarial when you learn from the mini-batch and generalize? I suspect that from the plots you've shown, most test examples are identified as non-adversarial but it looks like there are some adversarials that are very close to the tests (or even lower LID than some of the tests). If you try to get these removed, maybe you end up removing some tests. I'm wondering what the test-set rejection rate would be corresponding to the numbers in the table....

---

> > > > > > > > ### Author Response · Authors · 2017-12-23
> > > > > > > > **Recall rate**
> > > > > > > >
> > > > > > > > Thanks for your question.
> > > > > > > > On the CIFAR-10 dataset against the Opt L2 attack, our LID-based detector achieved: AUC: 98.94%, Accuracy: 95.49%, Precision: 92.98%, Recall: 93.54% (AUC score was reported in Table 1).

---

> > > > > ### Author Response · Authors · 2017-12-22
> > > > > **Re: BIM-b in Table 2?**
> > > > >
> > > > > Agree. We do find some interesting results and will address them in the next version.

---

### Public Comment · ~Aaron_Schumacher1 · 2018-05-04
**prior work**

I was surprised to see no mention of the 2004 NIPS paper by Levina and Bickel, "Maximum Likelihood Estimation of Intrinsic Dimension." Their equation 8 is the same as equation 4 in the paper of this page, except that they use k-1 instead of k. Levina and Bickel also point out that to get an unbiased estimator, the correct factor to use is k-2. Some quick experiments with constructed datasets support the correctness of Levina and Bickel (see gist, below). As far as I can tell, the Levina and Bickel paper is both prior to and more correct than the LID MLE formulation referenced in the paper here. The difference in factor shouldn't affect the results of the paper here, but it seems too soon to forget about work from 2004.

https://gist.github.com/ajschumacher/bdfecc8bdd48cad1b701b6cf4d647f4c

---

> ### Author Response · Authors · 2018-05-31
> **prior work**
>
> Thanks for the reference, we will add it to the paper in the arXiv revisions.

---

### Decision · Program_Chairs · 2018-01-29
**ICLR 2018 Conference Acceptance Decision**

**Decision:**

Accept (Oral)

**Comment:**

The paper characterizes the latent space of adversarial examples and introduces the concept of local intrinsic dimenstionality (LID). LID  can be used to detect adversaries as well build better attacks as it characterizes the space in which DNNs might be vulnerable. The experiments strongly support their claim.